# Lifestyle Entrepreneurship as a Vehicle for Leisure and Sustainable Tourism

**DOI:** 10.3390/ijerph20043241

**Published:** 2023-02-13

**Authors:** Miguel Duarte, Álvaro Dias, Bruno Sousa, Leandro Pereira

**Affiliations:** 1Management School, Instituto Superior de Gestão, Av. Marechal Craveiro Lopes, N.º2 A, 1700-284 Lisbon, Portugal; 2ISCTE, Instituto Universitário de Lisboa and BRU-Iscte—Business Research Unit (IBS), Av. das Forças Armadas, 1649-026 Lisbon, Portugal; 3School of Hospitality and Tourism (ESHT), Polytechnic Institute of Cávado and Ave (IPCA) and CiTUR, 4750-810 Barcelos, Portugal

**Keywords:** lifestyle entrepreneurship, leisure, sustainable tourism, rural areas

## Abstract

The subject of this research is related to sustainable tourism and its connection with lifestyle entrepreneurship. The Portuguese business fabric is formed by micro, small, and medium sized enterprises which have emerged in big numbers over the last years, mainly related directly and indirectly to the tourism industry. The discussed issue of this study is whether these companies are a vehicle for sustainable tourism in rural areas. Via a qualitative method, through a comparative case study of 11 businesses, the goal is to evaluate if the lifestyle entrepreneurship initiatives promote sustainable tourism in rural areas, identifying the specific business created and evaluating their growth toward the planned strategies and actions related to internal resources and capacity, as well as marketing. Lastly, the results present the plans made for growth according to the necessary balance among economic progress, environment, public health, and a social context. This study promotes decision tools for entrepreneurs and destination managers as to the practices to be adopted with the goal of sustainable development. Thus, in terms of ecological responsibility, the use of renewable energy through biomass is a very efficient practice because it both produces energy and reduces waste, since the energy production comes from plants and animal waste.

## 1. Introduction

Tourism has in recent years been an industry that exceeds all expectations in terms of worldwide projections. According to the World Tourism Organization, a specialized agency of the United Nations, the flow of international tourist arrivals grew worldwide by 4% in 2018, generating faster revenue growth than the world economy, making this industry a force not to be disregarded for the economic growth of nations, as well as for their development creating jobs promoting innovation and entrepreneurship [1]. Following the global panorama, the tourism industry in Portugal has an important weight in the national economy, with the consumption of this industry in Portuguese territory representing 14.6% of the gross domestic product. Furthermore, this is the economic activity that has more expression at the level of service exports, with a representation of 52.3%, in addition to representing 19.7% of total national exports [2]. As such, tourism has been in recent years the main driver of the Portuguese economy, being a strategic activity for Portugal in terms of attracting investment.

Therefore, the growth of activity generated by the tourism industry has grown along with business opportunities and the creation of companies to provide services to the sector, which in turn has awakened the entrepreneurial feeling in Portugal. According to what has already been reported by some media, the year 2018 was a record year in the opening and creation of new companies in Portugal, with 40% of them linked to tourism activities [3,4,5]. This fact combined with the strong growth of tourism activity in Portugal makes even more evident the need for the performance of activities in the tourism sector in strong awareness with sustainable development. Sustainable development is present in tourism activities at all their destinations of involvement.

The progress of these activities must be developed in balance with economic, environmental, and sociocultural development, in order to preserve and maintain the attractiveness of the destinations in which they are inserted, representing the key factors of attractiveness for their visiting customers. These principles apply to all types of destinations, whether mass tourism or any other segments and niches, such as rural tourism. As mentioned above, the exponential increase in business creation driven by the growth of tourism activity in Portugal has led to a huge number of new business openings. The Portuguese business fabric is mostly composed of micro, small, and medium enterprises, which represented 96.6%, 3.3%, and 0.5% of the total in 2018, respectively [6]. These 99.9% are companies that largely support the Portuguese economy and, consequently, tourism activity. This new wave of business creation can be seen as an entrepreneurial surge, carried out by individuals who have found opportunities and can be framed in the concept of lifestyle entrepreneurs, a concept that is scrutinized later as an inclusion variable of this study. In this sense, the theme of this research focuses on sustainable tourism, involving the entrepreneurial wave of lifestyle driven by the growing importance of tourism in economic activity, with a focus on this activity in rural space.

Furthermore, tourism businesses run by lifestyle entrepreneurs are found to be more sustainable when compared to bigger companies [7]. Sustainability issues are a major concern in the industry, and they have gained a central place in tourism research [8]. By focusing on financial and nonfinancial objectives, tourism entrepreneurs consistently follow sustainable goals related to the community and the environment [9]. Although these are important issues, research on sustainability in lifestyle entrepreneurship is still scarce [10]. As such, the aim of this manuscript is to evaluate if lifestyle entrepreneurship initiatives promote sustainable tourism in rural areas, identifying the specific businesses created and evaluating their growth toward the planned strategies and actions related to internal resources and capacity, as well as marketing. 

The question that drives this development is whether these entrepreneurial initiatives are a vehicle for the sustainable development of tourism in rural areas, with the aim of assessing whether these initiatives actually promote the sustainable development of tourism activity in rural areas. That said, lifestyle entrepreneurs are interviewed in relation to their businesses, assessing their attitudes and practices in terms of ecological responsibility, social responsibility, marketing, and innovation. The capacity of their existing internal resources, in terms of their operation, human resources, and tourism resources, is directly related to customer revisit as a performance indicator provided by the businesses.

The structure of this manuscript consists of four sections, followed by the conclusion. The first section has the purpose of presenting sustainable tourism, setting out the foundations on which it is based, as well as the main guiding bodies of the guidelines proposed for the sustainable development of tourism activity, both internationally and in Portugal. The second section introduces the concept of lifestyle entrepreneurship, involving tourism activity in particular. The third section serves to characterize the main research variables, namely, ecological responsibility, social responsibility, marketing, and innovation. The fourth section presents the methodology carried out to design the interview and data collection in order to present the results. The last section is dedicated to presenting the results and conclusions, along with the limitations of the study and proposals for future research.

## 2. Literature Review

### 2.1. Sustainable Tourism

Sustainable tourism is a topic with a very comprehensive body of literature, and it is always linked to the intrinsic root of sustainable development. The growing awareness of sustainable development is largely due to the World Commission on Environment and Development created by the United Nations in 1987, which developed the Brundtland report outlining guidelines for present development to be implemented without compromising the resources and needs of future generations [11]. With this, the triple bottom line concept gained prominence since sustainable development should be based on the balance of three fundamental pillars: economic, environmental, and social development [12]. Nevertheless, over the course of many decades, sustainability has gained more and more theoretical depth and acceptance; however, much of the literature questions the lack of methodology for its practical application [11]. On the other hand, more recently, new goals for sustainable development were set by the United Nations through the 2030 agenda for sustainable development. In this new elaboration, there is a new and more grounded awareness based on fighting poverty, preserving the environment, and maintaining economic prosperity [13]. As such, the 2030 agenda was elaborated by defining 17 goals to achieve 169 objectives for sustainable development. According to the World Tourism Organization, the tourism sector has the potential to contribute directly and indirectly to all the proposed goals, as particularly embedded in three of the 17 goals. The first is related to economic growth and the creation of decent work. The second is responsible consumption and production. Lastly, the third goal directly related to tourism activity is aquatic life [14]. Accordingly, sustainable tourism and entrepreneurship are related on the basis of these guidelines. According to [15], sustainable tourism in Portugal is progressively becoming both a trend and a concern. The development of sustainable practices is evident in business efforts (e.g., glamping) and in relation to the motivations of tourist demand (e.g., eco-tourism, green tourism, and slow tourism) in Portugal.

### 2.2. Lifestyle Entrepreneurship

Research linking entrepreneurship and tourism activity intertwines both concepts with small family businesses and lifestyle entrepreneurship. Nevertheless, the fact that small family businesses represent expressions of entrepreneurship serves as a debate in academic terms [16].

Authors [17] stated that small businesses in the tourism industry are generally designated as having a specific lifestyle orientation. Following the identification of this label, its definition is closely linked to the economic and social contexts in which certain businesses are created [18], leading to enormous complexity of its definition; however, it is linked to personal fulfillment, without a primary connection to conventional economic success, but with a new lifestyle that the entrepreneur desires as a change in their life, combined with basic economic subsistence and social integration. In more recent studies, the same author posits that the characteristics of small-scale entrepreneurs in the tourism sector differ from other economic activities. As an example, such a difference can be observed in the migratory movements to rural areas to meet the goals of changing lifestyles in large metropolises.

On the other hand, [19] proposed a framework for the study of entrepreneurship in the tourism and hospitality industry, involving the domains that influence the creation of business and the entrepreneurial process, as well as the results of entrepreneurial activity. This proposal is a consequence of weak theoretical development on the subject, despite good practical development. In this sense, the development of this research is based on this framework to enrich the research area in question. Lifestyle entrepreneurship has recently been debated in the academic community. For example, [20] developed a study in which the target population was lifestyle entrepreneurs who operate in Portugal and Spain. Firstly, this is one of the few empirical studies to research factors influencing innovation and entrepreneur self-efficacy for lifestyle entrepreneurs. From their point of view, on the basis of empirical evidence from Portugal and Spain, the authors were able to develop a model emphasizing the importance of these factors. Secondly, by exploring the relationship of the constructs mentioned above, their study expanded and updated the lifestyle entrepreneurship literature. Thirdly, to their best knowledge, this was the first study to analyze the construct of marshaling in tourism lifestyle entrepreneurship (in Portugal and Spain). By doing so, their study made an important contribution to both small businesses and (Portuguese and Spanish) destination competitiveness.

### 2.3. Ecological Responsibility

Ecological responsibility emerges as one of the fundamental factors for sustainable development through environmental sustainability [21]. Sustainability, in its broad sense, consists of ensuring the preservation of the evolution of future societies, keeping in mind their natural habitats, while considering economic and socioenvironmental actions and behaviors.

Ecological responsibility has gained prominence over time due to climate change and its global consequences. Therefore, the implementation of environmental sustainability is one of the key determining factors for the sustainable development goals set by the United Nations and adopted by 193 countries [22].

Ecological responsibility involves all practices and strategies put into action with the goal of environmental preservation and climate change mitigation. One of the practices is the choice of renewable energies to the detriment of conventional fossil fuels that are among the main emitters of carbon dioxide into the atmosphere. By renewable energy, we refer to energy sources that do not run out with use and are renewed by their own sources. There are several types of renewable energy:-Biomass that produces bioenergy from plants and animal waste;-Solar energy;-Wind energy;-Hydraulic energy;-Geothermal energy [23].

Strategies for waste management by promoting waste recovery for energy production, such as reuse by giving objects a new sense of use and recycling, also mirror ecological responsibility [22]. From a tourism perspective, on which this research is based, according to a study conducted by [24], the types of waste can be classified as waste residual, organic, and recyclable products. Of these types of waste presented, the practices for their reduction focus mostly on preventing food waste, reducing the unique use of plastics, and increasing separation and recycling.

On the other hand, ecological certification or obtaining an ecolabel, which proves good ecological practices, can be proof of ecological responsibility. According to [25], certification consists of voluntarily adopting a process that evaluates, audits, and gives guarantees of something that follows specific standards, labeling with a seal of recognition who meets those same standards. As far as tourism is concerned, ecoTourism has been a massive segment in recent decades, along with the creation of a panoply of certifications based on internationally institutionalized standards. However, this growing appearance of eco-certificates triggers some skepticism and criticism. Effectiveness and efficiency of international standards at the local level do not always have adequate relevance to the local needs where they are applied. The authors also mentioned the motivation to pursue such a certification. With respect to Scandinavian tourism companies, ref. [25] also concluded that certification in this area is low in number, identified by lifestyle entrepreneurs as a common point among companies with ecological certificates. In Portugal, ecological responsibility has been debated and developed. For example, ref. [26] developed a study on determiners in the consumer’s purchase decision process in Portuguese ecotourism contexts.

### 2.4. Social Responsibility

Increasingly, companies have a responsibility with regard to their management choices and attitudes toward and for society, as well as the expectations it has of them. Responsibility toward society is closely linked to environmental responsibility; thus, when addressing the topic of social responsibility, the environmental factors and good ecological practices are intrinsic aspects and points to be discussed. That said, corporate social responsibility can be defined by the set of attitudes, strategies, and management choices that a company makes with a view of achieving a balance among economic, social, and ecological sustainability. The balance of these three aspects is the triple bottom line approach that is the basis of sustainability.

According to [27], corporate social responsibility encompasses four correlated strands. First, the economic responsibility viewpoint is related to the economic objectives of producing goods and services according to the needs of society with the mission of making profits. Allied to this economic livelihood factor, there is a reciprocity of dependence, as companies need the members of society to whom products and services are provided, just as society needs companies for its development and satisfaction of needs. With economic responsibility comes legal responsibility, since the development of business activities presupposes compliance with the legal frameworks of the society in which the business operates. Ethical responsibility also appears as part of social responsibility, based on expectations, the relationship with society, and its social norms. Lastly, the authors mentioned a philanthropic aspect, which can be seen in the context of charitable actions or even actions where the company and entrepreneurs include society on a voluntary basis, promoting a sense of belonging on the part of entrepreneurs.

Authors [28] investigated the impact of cooperative social responsibility practices by large hotel chains. The study was conducted from the perspective of approaching employees. The impact of the companies’ actions on workers was investigated, in addition to how this can change their behavior in society, introducing new practices in society and changing some social norms. The conclusions revealed that companies must have social responsibility by adapting to the culture and social norms in which they operate, becoming a role model themselves as they also ensure their needs and wellbeing.

According to [29], studies on this topic mostly took into account large companies, while studies evaluating small and medium-sized companies are scarcer. This is primarily due to the visibility and media impact of large organizations in the world and society. However, according to the European Commission, 99% of all businesses in the European Union are small and medium-sized companies, thus highlighting the importance of having more studies contemplating these companies and evaluating their impact on sustainability actions. In Portugal, social responsibility has also been debated and developed. Portuguese companies have progressively understood that corporate social responsibility is a very important factor for future generations and for competitive differentiation. Consumers (Portuguese) increasingly value socially responsible (business) practices (externally and internally). As an example, we have the defense of sustainability, the fight against waste, the reduction in garbage and waste, and the fight against animal abandonment [30].

## 3. Methodology

### 3.1. Research Design

Aiming to evaluate whether lifestyle entrepreneurship initiatives promote sustainable development in rural tourism, this study was based on the observation of a qualitative comparative study of 11 companies operating in rural areas in the tourism sector.

The group of interviewees includes entrepreneurs who own and manage small-scale enterprises with a maximum of 12 rooms, located in the central region of the country and classified as rural hotels, country houses, residential tourism, or local accommodation, with accommodation as their main activity. The research was carried out through a comparative case study of 11 business units, taking into account the inclusion variable of lifestyle entrepreneurship. Accordingly, businesses developed by lifestyle entrepreneurs were considered in this study.

### 3.2. Data Collection and Sampling

The research and data collection were carried out through the elaboration of a semi-structured interview according to the topics inherent to the approach of the variables exposed in the literature review. Each interview took between 30 and 50 min. The aim was to relate and observe the different actions taken by each entity as a function of their impact on the performance of the stakeholders. The performance evaluation was conducted using the revisit rate for the years 2017, 2018, and 2019. The time period under analysis was that prior to the pandemic situation generated by the SARS-CoV-2 virus that plunged the tourism industry into a crisis of lack of demand. Not many studies have related guests’ attitudes and revisit intentions to whether hotels have sustainable or green practices in their management and operation. Data were captured and then transcribed. The transcripts were eventually translated into English because the dialogue occurred in Portuguese. To ensure reliability, we followed procedures used in previous tourism research (c.f. [31]). Since it is natural for respondents with diverse interests to deliver information in different ways when using a qualitative technique, we recruited five professors and researchers in tourism to respond to the questions as a pre-test prior to the official interviews (Table 1).

The number of interviews was not initially defined since we followed the approach proposed by [32] regarding the saturation point. Saturation is a continuous, cumulative judgment that one makes, which may never be attained; therefore, it is dependent on the analyst. In this case, saturation was reached after 10 cases; however, the researchers performed one more interview, which did not add significant information to the previous ones.

### 3.3. Instruments

Environmental concern has grown in recent years, and the environmental impact of the tourism and hotel industry is no exception. Therefore, in order to meet the growing demand for greener products, there has been an effort by the hotel industry to adopt green and environmentally friendly practices in order to improve their reputation, thereby bringing a competitive advantage with long-term effects on guest loyalty [32]. The choice of revisit rate as a performance variable is related to its link to customer satisfaction, suggesting that a customer’s experience at the hotel exceeded their expectations depending on the attitudes and practices of the entrepreneurs in the businesses they run. Accordingly, their impact on the customer brings about another visit or purchase of the services proposed, while also providing spontaneous word-of-mouth communication, attracting further customers [33].

Following the conclusion of the interviews, the qualitative data were gathered and evaluated to create themes using content analysis techniques. Subsequently, impartial raters were employed to create an environment for interaction to uncover themes and concepts. Due to the research paradigm’s subjective characteristics, the qualitative outcomes were coded, categorized, and thematized, turning the raw data into usable information. The data were analyzed considering the codes described in Table 2.

## 4. Results and Discussion

In this section, the results obtained from the interviews with the entrepreneurs are explored, showing the practices common to the players, as well as the aspects that differed or stood out. Ecological responsibility was common awareness among the interviewees; however, the practices and attitudes were not fully demonstrated:


*“Renewable energy is very important in ecological terms, and we aim to install solar panels in the near future to use as one of the energy sources in the house.”*
(Interviewee 3)

Similarly, interviewees 5 and 8 did not include renewable energy as an energy resource used in their businesses, although they intended to apply it in the long term. The use of renewable energy is a practices that best conveys the ecological responsibility of those involved. Solar energy using photovoltaic panels is the most common means used by most entrepreneurs for the production of energy to sustain the operation of their business, whether for general energy of the enterprises or for heating the water used by guests and employees. The use of biomass was also one of the energy resources adopted, especially for heat production and heating of the accommodation units under study. It should be noted that the use of biomass is both a waste reduction and an energy production strategy [24]. In line with [23], it appears that solar energy and biomass were the resources most commonly used as renewable energies, with no use of hydro, wind, or geothermal energy.

On the other hand, the policies and strategies for waste reduction were more diversified. A common practice observed was the appeal and awareness of all entrepreneurs toward customers who enjoyed the services provided and the localities in which the businesses in question were located. The strategy that stood out was interviewee 1’s implementation of a QR code system around the locality, appealing to the fight against waste, while advising good practices both in the unit and in the surrounding places of interest. Nevertheless, in line with [24], a reduction in residual waste, as well as the use of organic and recyclable products, was practiced. These practices were demonstrated through actions such as the use of tare glass water bottles (sent back to suppliers), the reuse of organic waste for composting and animal feed, the reduction in plastics by replacing them (e.g., with more environmentally friendly materials such as paper or textiles that allow reuse), the recovery of rainwater from roofs for irrigation and a cistern that uses the water for flushing toilets, the use of water reducers in showers and taps, the reuse of expired food products for home consumption, and the recycling of furniture, giving it a new life for decoration.

In terms of eco-certification, only two of the respondents had an eco-label, converging with [25] study in Scandinavia, where they observed a low number of certifications.


*“Certification is not the channel that brings me customers”.*
(Interviewee 1)

From the perspective of these entrepreneurs, ecological certification was not advantageous because it was not a differentiating factor or did not add value to the services provided by their businesses in attracting more customers.

The social responsibility practices and attitudes of the entrepreneurs converged with the aspects presented by [27]; however, none of them presented the philanthropic practices described by the author. The economic aspect was inherent to all interviewees due to their objective of obtaining profit in line with the localities in which they were located. These practices were translated by the employment of employees mostly from the municipalities in which they were located and by the use of local products for their operations, as mentioned by interviewee 8 (“We use mostly local and traditional products to help local producers, and they are the ones who give authenticity to our service”). This practice was considered to maintain, preserve, and continue to develop traditional activities of the locality that are important for the success of the entrepreneurs’ businesses and for the satisfaction of customers, exceeding their expectations through their experiences. Nevertheless, secondary products to the development of the main activity were also supplied by local companies, such as maintenance work or website development. In this way, many of these attitudes also translated into ethical responsibility, as they met local expectations in relation to society itself and the natural norms established by it. On the other hand, not all entrepreneurs established close relationships and partnerships with local entities, be they institutional, cultural, or economic, except from a legal responsibility aspect imposed by the regions in which they operated. When acting with social responsibility, entrepreneurs also aimed to have an impact on their employees by molding themselves to the localities in which they were inserted, similarly to that observed by [28].


*“When we arrived, we had to adapt to the way of life of the local people. They found it strange that we used rainwater to water the vegetable gardens, but after a while they started to do the same so that they wouldn’t use up the water from the wells.”*
(Interviewee 7)

Innovation in terms of distribution and sales channels was important for all entrepreneurs, especially in the online environment, with Booking.com standing out as the main sales channel.


*“Our sales are all online. We have tried several sites and we really want to develop ours to sell directly, but the top one is Booking.com”.*
(Interviewee 9)

There was, however, a constant intention on the part of all interviewees to seek new sales channels, and the bet on direct sales through the website itself was given priority. In line with [34], the interviewed entrepreneurs sought innovation through process improvement, as well as the development of new products and services offered, while maintaining their positioning and organizational mentality.


*“We try to innovate with new products combining partners’ activities with ours, not running away from our essence”.*
(Interviewee 10)

In terms of processes and products, there was concern when analyzing the opinions of customers through the main means of distribution and local partnerships to develop new products, indicating local partnerships as fundamental for innovation and to remain abreast of the market, as well as for the implementation of new ecological practices. Thus, the existence of a co-creation of value by collaborating with partners and customers was verified for the development of new products and services, thus enriching the product offered, in accordance with the statements of [34,35] that innovation is a complex system of interactions between various actors whether economic, social, or institutional.

The findings show that environmental and social responsibility, marketing and communication, and innovation had an impact on the rural lifestyle entrepreneurs. In other words, they were able to convert their concerns with the environment and the local community in which they developed their activity into valuable experiences for visitors. Simultaneously, this provided new narratives for communication and marketing activities. This finding is important and extends existing knowledge, since previous research was only dedicated to knowledge integration [37] and the importance of financial and nonfinancial objectives [38]. 

In line with [39], the digital presence of the businesses developed by these entrepreneurs was mostly translated through the use of social networks such as Instagram, Facebook, TripAdvisor, and Booking.com as the main strategy for communication, dissemination, and sale of their products and services.


*“Besides Booking.com, our online presence relies heavily on the social networks Facebook and Instagram, where we make many publications during the day, with the activities in our field, especially in high season”.*
(Interviewee 9)

In terms of communication, the planning of photo and video releases on this platform was daily; however, this planning tended to be organic to the extent that it coincided with the periods of greatest demand. On the other hand, even if the social networks were the main means of promotion and communication for these businesses, their use for the development of campaigns was not a common tool, with only one player applying this approach. All entrepreneurs considered their ecological and social position an important feature of their communication and image, while most of them did not see an ecological certification as a sales channel or driver. On the other hand, participation in national tourism fairs was only addressed in the offline environment. However, only one of the entrepreneurs had participated in an international fair, and participation was not a constant strategy.

The revisit rate or customer return rate was an indicator provided by the entrepreneurs. Only one of the entrepreneurs responded that, in 3 years, 40% of his customers returned to the enterprise, whereas eight interviewees indicated that their revisit rate was at 20%, and three claimed that their customer return rate was below 3%. With these data, we can see that the revisit rate was low (below 50% on average). On the other hand, all entrepreneurs indicated their quality of service and friendly welcome as the main factors for customers to return to their units, without referring to their sustainable development practices.

By emphasizing the joint contribution of the social and environmental responsibility and the marketing activities to the innovation generated, as well its impact on firm performance, this study broadened our understanding. Secondly, this study examined a particular group of entrepreneurs in the setting of rural areas, addressing a gap in the literature [40]. Despite acknowledgment of its significance for creativity in the rural environment [41], this study helps better understand the variables that may boost this capacity.

## 5. Conclusions

### 5.1. Theoretical Implications

According to the study framework proposed for entrepreneurship in tourism by [19], this study contributed to understanding the processes of entrepreneurial activity in terms of sustainable tourism and local development. On the other hand, in line with [17], who associated small tourism businesses with a certain lifestyle, we can conclude that this lifestyle followed a sustainable development aspect reflected in the management options with which they operationalized their businesses.

The study contributes to existing knowledge regarding tourism and sustainability. Small tourism businesses are not only associated with more sustainable practices but also play an important role in innovation dissemination within and across destinations [20]. By studying their sustainable approaches and activities, as well as the importance of the three pillars of sustainability in the development of more innovative firms, this study addresses the challenge posed by [20] regarding the necessity for researching small tourism firms, thus providing valuable insights into the processes leading to more sustainable tourism firms. To our best knowledge, this is the first study to fill this gap in small rural lifestyle-oriented tourism businesses. Furthermore, the findings also extend existing knowledge in relation to the development of more sustainable business models. For example, [40] discussed the role of acquiring and assimilating local knowledge for sustainability. This study adds to this knowledge of the influence of the social responsibility.

Furthermore, this research established a link to strategy theory. The results showed that local ties to the community enhanced the chance of knowing other entrepreneurs and stakeholders, contributing to achieving the entrepreneurs’ objectives more quickly. This finding provides a valuable contribution to tourism entrepreneurship by showing an important strategic strength, thus linking previous research relating business models (cf. [7]) and rural entrepreneurship [42] with strategic thinking. This is especially important for specific strategic theories such as resource-based view (cf. [43]).

Lastly, our findings expand place attachment theory. Our results revealed that local embeddedness not only plays an important strategic role, but also functions as a dynamic process as a result of the entrepreneurs’ behavior. By establishing long-standing ties to the community and local stakeholders, these entrepreneurs benefited from community-centered activities (e.g., local purchases or involvement in local social and commercial associations). More specifically, they gained access to their collaboration and knowledge, representing a basis for differentiated products and experiences.

### 5.2. Practical Implications

This study promoted decision tools for entrepreneurs and destination managers with respect to the practices to be adopted with the goal of sustainable development. Thus, in terms of ecological responsibility, the use of renewable energy through biomass is a very efficient practice because it both produces energy and reduces waste, since the energy production comes from plants and animal waste. On the other hand, making partnerships with local entities and local suppliers, maintaining constant communication of information, and exchanging value contribute to social responsibility and innovation in the market. Lastly, the digital area is fundamental to the success of entrepreneurs, both in terms of communication and in terms of distribution, necessitating presence in social networks such as Instagram and Facebook for promotion, as well as Booking.com for distribution and sales. The evaluation of sustainable practices related to performance indicators of the activities developed by the entrepreneurs is fundamental for the analysis of their impact on the businesses themselves.

### 5.3. Limitations and Future Research

This study, like many others, had certain limitations. Firstly, it was difficult to conduct interviews with local interviewers and interviewees. Secondly, the collection of the data related to this study was typically seen by entrepreneurs as confidential; therefore, the data provided are not very assertive. This did not allow direct conclusions to be drawn between the application of sustainable practices and the performance of the businesses developed by the entrepreneurs. Thirdly, the elaboration of studies related to lifestyle entrepreneurs is becoming more and more important, especially in the area of tourism, since this activity is one of the main drivers of economic development in Portugal. The analysis of their motivations, their relationship with sustainability, and their impacts on the localities in which they operate may help in the planning of investment in rural areas and in promoting the desertification of some of these areas.

## Figures and Tables

**Table 1 ijerph-20-03241-t001:** Sample characterization.

Entrepreneur Interviewed	Function/Position	Region	Main Activity	Legal Classification	Number of Employees	Number of Rooms
1	Owner and manager	Center Portugal	Housing	Local accommodation	1	6
2	Owner and manager	Center Portugal	Housing	Residential tourism	2	6
3	Owner and manager	Center Portugal	Housing	Rural hotel	5	12
4	Owner and manager	Center Portugal	Housing	Country house	2	8
5	Owner and manager	Center Portugal	Housing	Local accommodation	0	4
6	Owner and manager	Center Portugal	Housing	Country house	3	7
7	Owner and manager	Center Portugal	Housing	Local accommodation	0	5
8	Owner and manager	Center Portugal	Housing	Country house	3	9
9	Owner and manager	Center Portugal	Housing	Residential tourism	1	6
10	Owner and manager	Center Portugal	Housing	Local accommodation	1	3
11	Owner and manager	Center Portugal	Housing	Local accommodation	1	5

**Table 2 ijerph-20-03241-t002:** Study variables and inherent topics.

Variables	Topics	Authors
Environmental responsibility	→ Use of renewable energies → Ecological seal/certification → Waste reduction policies	- [23] - [22] - [24] - [25]
Social responsibility	→ Percentage of employees native to the locality → Local product use → Partnerships with local entities/companies for the development of the activity/product itself	- [27] - [28]
Innovation	→ Distribution/sales channels → Cocreation of value by collaborating with partners and customers for the development of new products/services	- [34] - [35]
Intensity of marketing	→ Use of social networks for online campaigns → Participation in national and international fairs → Digital presence (website, TripAdvisor, and Booking.com)	- [36]
Return/review rate for the last 3 years (2017/2018/2019)	→ To quantify repeat customer visits and their annual percentage in the last 3 years.	- [33] - [32]

## Data Availability

Data are available upon reasonable request to the corresponding author.

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
