# Peer review of "Lifestyle Entrepreneurship as a Vehicle for Leisure and Sustainable Tourism"

_ijerph, 2023, doi:10.3390/ijerph20043241_

Round 1

Reviewer 1 Report

Dear authors, it was a pleasure to have reviewed your paper. Below you will find the changes to be made:

1.      In the abstract, please include the main outcomes in the last sentence, not all of them.

2.      After chapter 2 there is no chapter 3, and it goes directly to chapter 4. So, at the last paragraph of the introduction, please check again what items are included in each chapter and re-write it.

3.      I understand that the paper is focused on Portugal. In the literature review kindly include information about Portugal at the end of each section. The concept is to talk about e.g. sustainable tourism in general (which you already did) and right after, about the sustainable tourism in Portugal. By doing this, you are giving the readers information not only about sustainable tourism in general but in Portugal as well.

4.      There is no “results” section in your paper. I understand that you have incorporated your results in the “discussion” section, but this is something different. In the results, you should present your findings only and, in the discussion, you compare your findings with those of other papers. So, kindly add a section “results” or “findings” after the "methodology" section and before the "discussion" section.

Thank you

Author Response

Authors’ Response 

Manuscript ID: ijerph-2197531 

Title: Lifestyle Entrepreneurship as a Vehicle for Leisure and Sustainable 
Tourism 

Thank you very much for your previous feedback. We feel now very comfortable,  reinforced, and confident by your extremely positive feedback, thank you very much indeed! Regarding the comments received in the first round of reviews, we also appreciate them and took them into consideration modifying the manuscript accordingly. The updated manuscript includes changes which are highlighted in yellow. Below you can find our response to the comments and suggestions made by the reviewers. 

REVIWER 1 

Dear authors, it was a pleasure to have reviewed your paper. Below you will find the changes to be made:
1. In the abstract, please include the main outcomes in the last sentence, not all of them. 
R: checked and corrected accordingly. 
The abstract’s last sentence is now written this way: Finally, the outcome will present the plans made for growth according to the necessary balance between economic progress, environment, public health and a social context. This study promotes decision tools for entrepreneurs and destination managers as to the practices to be adopted with the goal of sustainable development. Thus, in terms of ecological responsibility, the use of renewable energy through biomass is a very efficient practice because it both produces energy and reduces waste, since the energy production comes from plants and animal waste.

2. After chapter 2 there is no chapter 3, and it goes directly to chapter 4. So, at the last paragraph of the introduction, please check again what items are included in each chapter and re-write it. 
R: Thank you so much. Checked and corrected accordingly. 
The sentence is now stated like this: The structure of this manuscript will consist of five parts followed by the conclusion. The first part has the purpose of presenting sustainable tourism, setting out the foundations on which it is based and the main guiding bodies of the guidelines proposed for the sustainable development of tourism activity, both internationally and in Portugal. The second part introduces the concept of lifestyle entrepreneurship, involving it with tourism activity in specific. The third part will serve to characterize the main research variables, namely ecological responsibility, social responsibility, marketing and innovation. The next section will present the methodology 
carried out to design the interview and data collection in order to present the results. The last part will be dedicated to the presentation of results, discussion and conclusion where the limitations and proposals for future research will be presented. 

3. I understand that the paper is focused on Portugal. In the literature review kindly include information about Portugal at the end of each section. The concept is to talk about e.g. sustainable tourism in general (which you already did) and right after, about the sustainable tourism in Portugal. By doing this, you are giving the readers information not only about sustainable tourism in general but in Portugal as well. 
R: Thank you so much. Checked and corrected accordingly. 
The authors agree with this comment. Content about Portugal (specifically) was 
developed at the end of each part of the literature review. Thank you so much. We now added this information: According to Fernandes et al. (2021), sustainable tourism in Portugal is progressively a trend and a concern. The development of sustainable practices is evident in business efforts (e.g. glamping) and in relation to the motivations of tourist demand (e.g. eco-tourism, green tourism, slow tourism) in Portugal. Lifestyle Entrepreneurship has recently been debated in the academic community. For example, Dias et al (2022) developed a study in which the target population of their research is 
lifestyle entrepreneurs who operate in Portugal and Spain. First, it is one of the few empirical studies to research factors influencing innovation and entrepreneur selfefficacy on lifestyle entrepreneurs. From their point of view, and based on empirical evidence from Portugal and Spain, the authors were able to develop a model in which is emphasized the importance of those factors. Second, by exploring the relationship of the constructs mentioned above, their study expands and provides an update regarding the lifestyle entrepreneurship literature. Third, to their best knowledge, this is the first study to analyze the construct of marshaling in tourism lifestyle entrepreneurship (in Portugal 
and Spain). By doing so, their study makes an important contribution both for small businesses and (Portuguese and Spanish) destination competitiveness. Also in Portugal, Social Responsibility has been debated and developed. Portuguese companies progressively understand that corporate social responsibility is a very important factor for  future generations and for competitive differentiation. Consumers (Portuguese) value (increasingly) socially responsible (business) practices (externally and internally). As an example, we have: the defense of sustainability, the fight against waste, the reduction of 
garbage and waste, the fight against animal abandonment (Soares & Sousa, 2022).

4. There is no “results” section in your paper. I understand that you have incorporated your results in the “discussion” section, but this is something different. In the results, you should present your findings only and, in the discussion, you compare your findings with those of other papers. So, kindly add a section “results” or “findings” after the "methodology" section and before the "discussion" section. 

R: Thank you so much. Checked and corrected accordingly. In this commentary, we have made a new adjustment in the chapters / sections in order to reconcile the results / findings and the discussion / reflection of the authors. We recognize that it is difficult to separate the two parts. In this sense, we reflected on "results" and "discussion" with some consistency.

Reviewer 2 Report

Thank you for opportunity reviewing paper titled “ Lifestyle Entrepreneurship as a Vehicle for Leisure and Sustainable Tourism” which I read with interest. The paper is interested; however, there are some concerns that required authors to address as follows:

1/ After I read one round of your paper, I can’t find a very clear of your study objective. This lead me to read for the second and third time. 

2/Author failed to clarify the research gap?

2/ Again go to discussion why we need this study in terms of theoretical and practical implications? 

3/ I think the last paragraph of your introduction part is too much and not necessary. Author should focus on “so what” of your study. 

4/ Literature review is quite developed but the authors just stop the show without giving audience with take home massage. Author failed to proof what the literature already exists and what do not exist? This make your study more interesting to conduct. 

5/ Author need to justify why on 11 companies?

6/ How the author conduct their coding? Missing again?

7/What kind of technique author apply for data analysis?

8/How the author guarantee the result reliability? This is very important for qualitative studies. 

9/ I think a clear code book procedure might help the reader to understand more. But here is missing again. 

10.The biggest issue for this paper is that the findings section of the paper is missing. I feel like the authors pay less attention on his or her study? Qualitative study relies heavy on descriptive results which make the reader understand more on respondent voice. 

11/ I do understand that you merge your finding with your discussion but that is not the study main findings. 

12/ without proofing your trustworthiness outputs, I can’t trust your discussion. It mean this paper is misleading the information. 

Author might take a look at the following paper, they are very careful on descriptive findings but good enough to convince the readers. 

Sann, R., Lai, P. C., & Chen, C. T. (2022). Crisis Adaptation in a Thai Community-Based Tourism Setting during the COVID-19 Pandemic: A Qualitative Phenomenological Approach. Sustainability15(1), 340.

Or Creswell, J. W. (2007). Five qualitative approaches to inquiry. Qualitative inquiry and research design: Choosing among five approaches2, 53-80.

13/ Finally, your conclusion, author failed to clarify what is the study theorectical and practical implications. This make the paper less contribution to the body of tourism and leisure knowledge. 

14, last but not least, limitation and further study recommendation is missing again . 

Hope it might help. 

Author Response

Thank you very much for your previous feedback. We feel now very comfortable, reinforced, and confident by your extremely positive feedback, thank you very much indeed! Regarding the comments received in the first round of reviews, we also appreciate them and took them into consideration modifying the manuscript accordingly. The updated manuscript includes changes which are highlighted in yellow. Below you can find our response to the comments and suggestions made by the reviewers. 

REVIWER 2 
Thank you for opportunity reviewing paper titled “ Lifestyle Entrepreneurship as a Vehicle for Leisure and Sustainable Tourism” which I read with interest. The paper is interested; however, there are some concerns that required authors to address as follows: 

1/ After I read one round of your paper, I can’t find a very clear of your study objective. This lead me to read for the second and third time. 
R: We agree with the reviewer. We now state more clearly the study objective In the 5th paragraph of the introduction: “The aim of this manuscript is to evaluate if the lifestyle entrepreneurs’ initiatives promote sustainable tourism in rural areas, identifying the specific business created, evaluating their growth towards the planned strategies and actions related to internal resources and capacity, as well as in relation to marketing. The topic of this research will focus on sustainable tourism involving it with the entrepreneurial wave of lifestyle driven by the growing importance of Tourism in economic activity, with the focus of this activity in rural space.

”2/Author failed to clarify the research gap? 
R: Thank you for pointing this out. We now added the reference to existing research in the field and specified what is missing in that research. This can be found in the 4th paragraph of the introduction. 
2/ Again go to discussion why we need this study in terms of theoretical and practical implications? 

R: Thank you for pointing this out. The discussion was revised according to the 
recommendations below. As such, the section was extended and revised to provide effective contributes besides existing presentation of the results. 

3/ I think the last paragraph of your introduction part is too much and not necessary. Author should focus on “so what” of your study. 
R: Thank you so much. Checked and corrected accordingly. From an interdisciplinary perspective, this manuscript presents insights for lifestyle entrepreneurship and sustainable tourism. The outcome will present the plans made for growth according to the necessary balance between economic progress, environment, public health and a social context. This study promotes decision tools for entrepreneurs and destination managers as to the practices to be adopted with the goal of sustainable development. Thus, in terms 
of ecological responsibility, the use of renewable energy through biomass is a very efficient practice because it both produces energy and reduces waste, since the energy production comes from plants and animal waste.

4/ Literature review is quite developed but the authors just stop the show without giving audience with take home massage. Author failed to proof what the literature already exists and what do not exist? This make your study more interesting to conduct. 
R: Thank you for pointing this out. We extended the literature review for a better presentation of existing research. Furthermore, the introduction was expanded to provide a presentation of current and needed research. 

5/ Author need to justify why on 11 companies? 
R: Thank you for pointing this out. We followed a data saturation approach which lead to 11 cases. The explanation for this procedure is now presented in the 4th paragraph of the methodology. It states like this: The initial number of interviews was not initially defined since we followed the approach proposed by Saunders et al. (2018) regarding the saturation point. This means that saturation is a continuous, cumulative judgment that one makes, and may never fully complete, therefore it can only be a question of the analyst's choice (Saunders et al. 2018). In this case, the saturation was reached at the 10th case, however, the researchers performed one more interview, which didn’t add significant information to the previous ones. 

6/ How the author conduct their coding? Missing again? And 7/What kind of technique author apply for data analysis? 
R: Thank for these important questions. We agree that important information is missing. 
We now provide detail regarding the content analysis conducted for this study. As such in the 6th paragraph we added this: “Following the conclusion of the interviews, the qualitative data were gathered and evaluated to create themes based on content analysis techniques. Following that, the research employed impartial raters to create an environment for interaction to uncover themes and concepts. Due to the phenomenal research paradigm's subjective character, the qualitative outcomes. The qualitative data were then coded, categorized, and themified, turning the raw data into usable information. The data was analyzed considering the codes described in table 2.” 

8/How the author guarantee the result reliability? This is very important for qualitative studies. 
R: Thank you for pointing this out. We followed your recommendation of example (Sann et al., 2022), indicated below, and developed our procedures according to that very well written paper. We flowed similar procedures, which are now described in the 3rd paragraph of the methodology, stating like this: “The research and data collection were carried out based on the elaboration of a semi-structured interview according to the topics inherent to the approach of the var-iables exposed above in the literature review. Each interview took between 30 and 50 minutes. The aim is to relate and observe the different actions taken by each entity with the results they have on the performance of the stakeholders. The performance evaluation is done using the performance variable - revisit rate for the years 2017, 2018 and 2019. There are not 
many studies that relate guests' attitudes and revisit in-tentions to whether hotels have sustainable or green practices in their management and operation. Data was captured and afterwards transcription was done. The tran-scripts were eventually translated into English because the dialogue took occurred in Portuguese. To ensure the reliability we followed procedures used in previous tourism research (c.f. Sann, Lai & Chen, 2022) since it is natural for respondents with diverse interests to deliver information in different ways when using a qualitative technique. As such, we recruited five professors and researchers in tourism to respond to the questions as a pre-test prior to the official interviews.” 

9/ I think a clear code book procedure might help the reader to understand more. But here is missing again. 
R: We hope have responded to this question in the response to #6. 
10.The biggest issue for this paper is that the findings section of the paper is missing. I feel like the authors pay less attention on his or her study? Qualitative study relies heavy on descriptive results which make the reader understand more on respondent voice. 
R: we agree with the reviewer. The results section was revised for a more detailed presentation of the results, including we included an extract of the conversations in the first person for a better illustration. 

11/ I do understand that you merge your finding with your discussion but that is not the study main findings. 
R: we agree with the reviewer. The discussion section was extended to include a more clear description of the contributions and to indicate specifically where the study advances existing knowledge in this field. As such, we believe that the section can be renamed ‘Results and Discussion’. 

12/ without proofing your trustworthiness outputs, I can’t trust your discussion. It mean this paper is misleading the information. Author might take a look at the following paper, they are very careful on descriptive findings but good enough to convince the readers. 

Sann, R., Lai, P. C., & Chen, C. T. (2022). Crisis Adaptation in a Thai CommunityBased Tourism Setting during the COVID-19 Pandemic: A Qualitative  Phenomenological Approach. Sustainability, 15(1), 340.

Or

Creswell, J. W. (2007). Five qualitative approaches to inquiry. Qualitative inquiry 
and research design: Choosing among five approaches, 2, 53-80.

R: Thank you for these valuable recommendations. We read carefully both documents and applied to our paper. We also used and cited the document from Sann et al. (2022) to exemplify these procedures in the tourism field. 
13/ Finally, your conclusion, author failed to clarify what is the study theorectical and practical implications. This make the paper less contribution to the body of tourism and leisure knowledge. 
R: We agree with the reviewer. We now separated the conclusions in three sub-sections. 
One is dedicated to the theoretical implications which were revised and extended for a more clear presentation of the effective contributions. The other corresponds to the practical contributions. Also revised. And we add a 3rd section to respond to your next suggestion.

14, last but not least, limitation and further study recommendation is missing again . 
R: Thank you for pointing this out. We now included a new section to present the limitations and future research avenues. 
Hope it might help.
R: Thank you for all your valuable recommendations. We hope they are correctly 
addressed.

Round 2

Reviewer 1 Report

Dear authors,

Thank you very much for the revision of your paper.

There is no other comment to be made.

Author Response

The authors would like to thank reviewer 1 for the positive and favorable assessment of our manuscript. The team of authors appreciates all comments and suggestions from the first round, which allowed us to improve the manuscript for favorable evaluation. Thank you so much.

Reviewer 2 Report

Thank you for your revision. I can see much improvement from the previous comments. However, there are still some minor comments that I would like authors to address to make this manuscript more suitable for this journal as well as to the reader. 

1- authors should separate the research method into sub-section or sub-title so it is easy to read. 

2- I am still did not see much improvement on your conclusion; besides, most of the content are from the previous version.

If the authors can improve the abovementioned points, then I think it is suitable for publication. 

Author Response

Thank you for your revision. I can see much improvement from the previous comments. However, there are still some minor comments that I would like authors to address to make this manuscript more suitable for this journal as well as to the reader. 

R: Thank you for the supportive comment. We hope having addressed all you recommendations.

1- authors should separate the research method into sub-section or sub-title so it is easy to read. 

R: Thank you for the recommendation. We divided the methodology section in three sub-sections.

2- I am still did not see much improvement on your conclusion; besides, most of the content are from the previous version.

R: Although several changes were made in the previous revision, it is not considered sufficient. To address this issue, we worked on the conclusions to provide a more detailed description of the contributions.
